# Generating More Hydroelecticity While Ensuring the Safety: Resilience Assessment Study for Bukhangang Watershed in South Korea

**Dong Hyun Kim** [1], **Taesam Lee** [2], **Hong-Joon Shin** [3] **and Seung Oh Lee** [1,*]

1 Department of Civil and Environmental Engineering, Hongik University, 94 Wausan-ro, Mapo-gu, Seoul 04066, Korea; uou543@gmail.com

2 Department of Civil Engineering, ERI, Gyeongsang National University, 501 Jinju-daero, Jinju 52828, Korea; tae3lee@gnu.ac.kr

3 Hydropower Research and Training Center, Korea Hydro & Nuclear Power Co., Ltd., 1655 Bulguk-ro, Gyeongju 38120, Korea; h.j.shin@khnp.co.kr

* Correspondence: seungoh.lee@hongik.ac.kr; Tel./Fax: +82-02-325-2332

**Abstract:** The recent integrated water management policy and carbon-neutral policy can be seen as a turning point that changed the major frameworks of water resource policy and energy policy in the world. Values of hydropower reservoirs, directly related to both policies, should be re-evaluated in terms of resilience. In the past, hydropower reservoirs in Korea have contributed both to flood control and to generating electricity when operating dams within the limited water level during flood seasons. Under such limited operations, the power loss would be inevitable. Therefore, in this study, the concept of resilience was introduced for application to the operation of the hydropower reservoir to minimize such power loss. Also, the framework was able to be used for evaluating power generation performance when setting the target function to the maximization of electricity sale profit. HEC-5 was used for deriving the optimal operation rule, and the scenario was established by referring to the procedure of the general multiple-reservoir operation plan in Korea. As a result of application to the proposed framework, the operation rule that produces the maximum amount of electricity sales was presented, and it was confirmed that flood control and water usage performance could additionally be evaluated. When comparing the past data with optimal operation results for the period 2006~2013, it was found that the resilient operation increased by about 19.83% in terms of electricity generation. In the near future, if various scenarios are added and economic analysis is accompanied, it will be able to judge the best economic effects and the least opportunity costs.

**Keywords:** resilience; hydroelectricity; reservoir; Hangang watershed; dam safety; power generation





## 1. Introduction

In Korea, hydropower reservoirs have been constructed, operated, and managed for about 90 years, starting with the Unam hydropower plant in 1931. Until the 1960s, when electricity was scarce, the hydropower reservoir was operated as a baseload power source for the power system. Since the 1980s, the hydroelectric field has been developed by inventing new sources, such as pumped-water power plants, and until recently, has played a role as a source of peak load responsible for power quality. The hydropower reservoir plays an important role as a power source in case of emergency, such as a sudden power outage, as it enables rapid electricity production due to its short operating and downtime. Although it faithfully performed its role for power generation, it has been given a role beyond power generation due to its specificity of using water resources as a power source. In other words, it can be seen to be in line with the value of water resources along with power generation.

In Korea, there has been a continuous conflict between the management and distribution of water resources because there is a large seasonal variation in the amount of

precipitation and the amount of water resources varies by locality. Recently, as the frequency of disasters such as floods and droughts due to climate change and when climate variability increases, the efficiency of water resource use is being emphasized more [1]. The role of several hydropower reservoirs in the Bukhangang watershed, the largest one in Korea, cannot be ignored in terms of water resources. In addition, it can be seen that the hydropower reservoir occupies an important position in terms of water resource management when considering the recent domestic policy stance, and its value needs to be evaluated anew.

Starting with the revision of the Government Organization Act in June 2018, as the Basic Water Management Act and the Water Technology Industry Act were enacted and amended, the task of integrated management of water quantity and water quality was integrated into the Ministry of Environment. This reorganization means that Korea's water management policy has been converted to water quality and environmental management, and it is intended to manage water quantity, water quality, and response to water disasters in a unified system. The hydropower reservoir located in the Bukhangang watershed is also included in the integrated water management system and contributes greatly to major key achievements. It is necessary to clearly present the role of the hydropower reservoir in this policy framework. Accordingly, in June 2020, Korea Hydro & Nuclear Power announced the multi-purpose use of the hydropower reservoir, emphasizing the role of the hydropower reservoir manager in watershed management [2].

In July 2020, the Korean government proposed a national project "Korean New Deal," and the most notable among them is the "Green New Deal" policy that promotes sustainable growth. Hydroelectric power generation is closely related to both the green energy sector of the Korean Green New Deal and the infrastructure green transition sector of the establishment of a clean and safe water management system. The role of hydroelectric power corresponds to the Korean version of the Green New Deal. Meanwhile, in October 2020, the government declared carbon neutrality by 2050, replacing coal power with renewable energy. Carbon neutrality means that the amount of carbon emitted is equal to the amount of carbon absorbed so that the net carbon emission becomes zero. In December 2020, a carbon-neutral promotion strategy was prepared, and one of the three major policies is to switch the main energy source from fossil fuels to renewable energy. Hydroelectric power generation becomes a representative new and renewable energy and is expected to become a necessary energy source to achieve the goal of 2050 carbon neutrality.

As such, although the framework of a new development opportunity for the hydroelectric industry has been prepared for the policy base, difficulties are occurring in not being able to follow the policy base due to social disputes and the absence of objective value evaluation. Efficient use of energy resources is expected to be important for the goal of carbon neutrality, but in the water resource sector, the use efficiency is low compared to the level of the established infrastructure. According to statistics, the average use rate of river water compared to the permitted amount of river water from 2013 to 2017 was about 60.9% in Korea [3]. In addition, there is a lot of room for technical and institutional improvement in terms of dam operation. In the technical aspect of dam operation, various optimization methods such as linear programming (LP), dynamic programming (DP), and stochastic dynamic programming (SDP) have been studied. In most cases, operating rules are presented [4]. Therefore, it is not suitable for application to hydropower reservoirs for which power generation is the main purpose. However, hydropower reservoirs are required to be operated in consideration of water supply and flood control to respond to water disasters while giving priority to power generation. Such a change in operating conditions increases the need for a method for operating a hydropower reservoir for various purposes and a method for evaluating it.

Therefore, in this study, the concept of resilience is introduced to suggest an optimal operation method for hydropower reservoirs. Resilience in the field of engineering generally refers to the restoration of a system to its original state after a disturbance has occurred. At this time, resilience is defined as the degree and time of recovery [5]. This definition

is applicable to the generating capacity of a hydropower reservoir. Power generation is a function of water level and quantity, but water level and quantity are in inverse proportion to each other. It is necessary to maintain an appropriate water level and secure the quantity to maximize the amount of power generation. However, for the function of water supply and flood control, discharge must be performed, so the water level is lowered, and it takes time to recover to an appropriate water level for power generation. This concept can be substituted for resilience. Therefore, in this study, the definition and evaluation methodology for the resilience of hydropower reservoirs are presented.

The concept of resilience was first used in the field of ecology, and it was defined as an ecosystem restored to an equilibrium state after losing its original function due to internal and external disturbances [6]. Since then, the concept of resilience has been established in various fields according to the purpose of each field. Walker et al. (2004) also suggested three characteristics (resilience, adaptability, transformability) and their relationship to explain the Social Ecological System (SES) [7]. Also, some researchers have explained resilience by dividing it into ecological resilience and engineering resilience [8]. Ecological resilience is defined as the amount of disturbance that can be absorbed before a fundamental change in system structure and function occurs, and multistable states can be defined. It pays attention to persistence, change, and unpredictability [5]. On the other hand, engineering resilience was defined as resistance to disturbance and speed of return to the equilibrium. It is defined as a single equilibrium point. It focuses on the efficiency, constancy, and maintenance of a single stable state of system [9]. Therefore, engineering resilience is defined as robustness, which indicates the magnitude of resistance for system preservation, and rapidity, which is the recovery time required to replace damage.

A representative example of engineering resilience has been applied to infrastructure. NIAC (2009) defines it as the ability to reduce the size and duration of disasters and analyzes that highly resilient infrastructure reduces the damage and scale of various disasters and minimizes losses by reducing the time required for recovery [10]. In addition, it was used for various facilities such as power transmission facilities and water supply facilities in the field of disasters, and the concept of resilience according to the characteristics of each facility was established. In Korea, the establishment of structural and non-structural alternatives for disaster response was explained as the concept of resilience [11]. They argued that structural alternatives should be established to reduce damage caused by disasters, and non-structural alternatives to reduce disaster recovery time should be established. There is also a case of applying resilience to multi-purpose dams and agricultural reservoirs. Kim et al. (2014) evaluated the flood control function of multi-purpose dams by introducing the concept of resilience and presented a method to evaluate alternatives for strengthening the safety of dams [12]. After that, Park et al. (2018) derived a drought water supply plan for each scenario considering the resilience for Lake Naju [13]. Kim et al. (2021) introduced the concept of resilience to evaluate the power generation capacity of hydropower reservoirs and suggested a method of maximizing the power generation [14].

## 2. Material and Methods

### 2.1. Resilience in Hydroelectricity Dam

Bruneau et al. (2003) defined the total loss of system in terms of time and functional level to express various attributes of resilience as a single value [15]. Since this definition expresses resilience as a single value, the comparative advantage of each scenario can be easily identified. In this study, the resilience of a hydropower reservoir using this analytical definition was applied [14]. It was defined based on engineering resilience, which is approached by focusing on the recovery time required to repair the damage done to the original properties [8]. Since the main purpose of the hydropower reservoir is to generate electricity, the components of robustness and rapidity were derived. Robustness was defined as the water level of the hydropower reservoir, and rapidity was defined as the time required to recover to an appropriate water level for power generation (Figure 1). The generation of electricity is determined by the effective head and the amount of outflow

from the reservoir. Thus, it is necessary to keep the water level high to increase the effective head and the amount of outflow. However, since the water resources stored in the reservoir are limited and the effective head decreases when the outflow for generation is increased, it is essential to properly maintain them. Since the water level is determined by the inflow and outflow, the function of the power generation system is represented by the dam water level. Therefore, a resilience triangle was defined with the vertical axis as the dam water level as shown in Figure 1a. With this conceptual approach, area B surrounded by the dotted line in Figure 1 is defined as the resilience and area A is defined as the total loss. The equation of the hydropower resilience is as follows.

$$R = \int_{t_a}^{t_b} W(t) - W_{lowest} dt \begin{cases} W_d(t) & t_0 < t < t_a \\ W_r(t) & t_a < t < t_b \end{cases} \tag{1}$$

where, $R$ is the resilience. $t_a$ is the point at which recovery begins. $t_b$ is the point at which recovery is complete. $t_0$ is the point at which the loss occurred. $W(t)$ is the water level function with time. $W_d(t)$ is the function of operation. $W_r(t)$ is the recovery function. $W_{lowest}$ is the low water level of each reservoir.

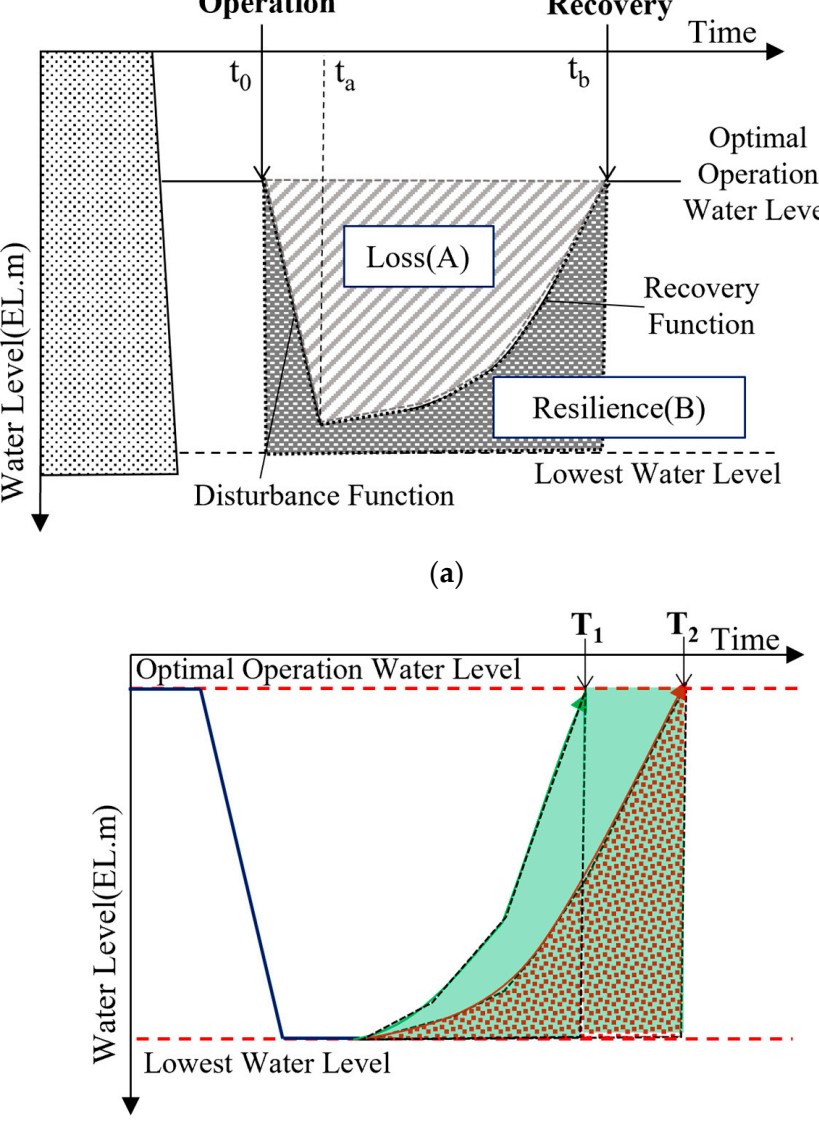

(**a**)

(**b**)

**Figure 1.** *Cont.*

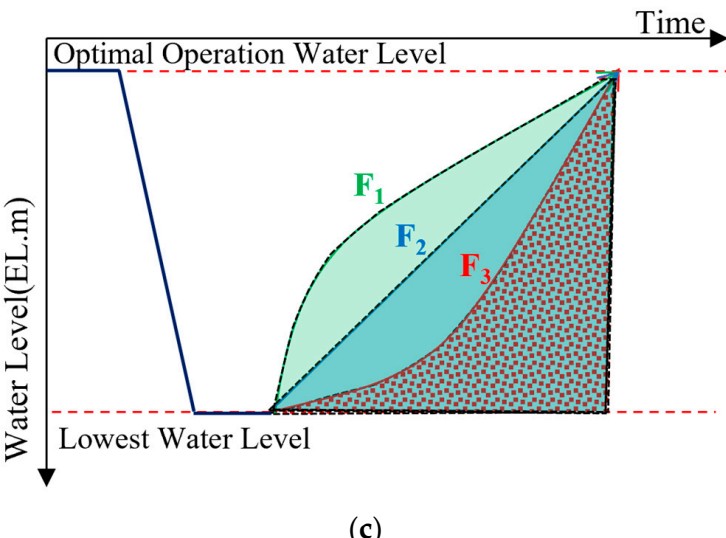

**(c)**

**Figure 1.** Resilience of Hydroelectric Dam. (**a**) Concept of Resilience; (**b**) Resilience with recovery time; (**c**) Resilience with recovery function.

Examples of applying this concept are shown in Figure 1b,c. Figure 1b shows an example of a different recovery time. $T_1$, which has a faster recovery time at the same time interval, has greater resilience than $T_2$ ($R_1 > R_2$). This is because the case of $T_1$ operates with a higher effective head, so the amount of power generation is higher. Figure 1c shows the case where the recovery time is the same, but the recovery function is different, and it can be said that $F_1$ has greater resilience than $F_2$ and $F_3$ ($R_1 > R_2 > R_3$). In both examples, it can be said that the larger R is, the better the performance at the hydropower reservoir. Using this concept of resilience, it is possible to present a methodology for evaluating the power generation performance of reservoirs.

*2.2. Dam Operation Modeling*

Representative models currently in use are HEC-5, HEC-ResSim, HYDROSIM, and MIKEBASIN. The HEC-5 used in the operation simulation of the dam in this study was initially developed to simulate flood control, but through continuous improvement, it has reached the most recent version, Version 8.0 (1998.10). This model can suggest the optimal operation plan of the dam group by maximally satisfying various purposes such as hydropower generation, water supply, and flood control under various boundary conditions in a system composed of several dams and control points. HEC-5 is configured to harmoniously maintain the water system while satisfying the constraints of each dam and the specified flow at the downstream control point. The priority of discharge by dam is determined by the index level. The concept of an equivalent reservoir is applied to a group of dams configured in series or parallel to determine the discharge priority. All dams in the system are operated to maintain the same index level. The discharge priority between dams is configured to discharge from the dam with the highest index level at every simulation time using the relationship with water level and storage capacity. HEC-5 is based on calculating the water level of the reservoir and the flow rate downstream. While securing a space to control floods, users can set target values for discharge amount, river maintenance flow, and hydraulic energy. In addition, seasonal rule curves and operating guide levels can be specified in HEC-5. Several optional hydrological flood routing methods are available. It is possible to calculate river maintenance flow, hydraulic energy, and annual flood damage, including calculating the constant guarantee amount for various water intakes. HEC-5 has a limitation in that it has not improved any more due to the development of HEC-ResSim, a later model. However, in this study, it is necessary to repeatedly calculate power generation and analyze water balance for reservoir operation scenarios, and various functions of HEC-ResSim specialized for education and real-time operation are

not required. Therefore, HEC-5, which can perform scenario-based iterative models, was selected as the dam simulation operation program. In particular, the selection of HEC-5 is inevitable because the US Army Corps does not provide COM interface information for HEC-ResSim.

The governing equation of reservoir flood routing in HEC-5 is based on the continuous equation and mainly uses the plus method or the modified plus method. However, if the reservoir is controlled by the gate, it is determined by the determination method of outflow discharge. For all dams in this study, outflow discharge is determined by the operation of the gate. The dam water level can be controlled by the outflow discharge of the reservoir. Therefore, the performance of the HEC-5 calculation process was evaluated using the observed inflow, discharge, and water level-capacity curves. The target of evaluation is Hwacheon Dam. The evaluation period is from 2011 to 2020. The water level-capacity curve made by 2015 was used. In Figure 2, it was confirmed that the observed data of Hwacheon Dam and the simulation results of HEC-5 were generally similar. The statistical correlation between the two data was 0.9437 for NSE and 1.15 m for RMSE. Since the error is not large and shows a consistent trend, the calculation of HEC-5 is evaluated as appropriate.

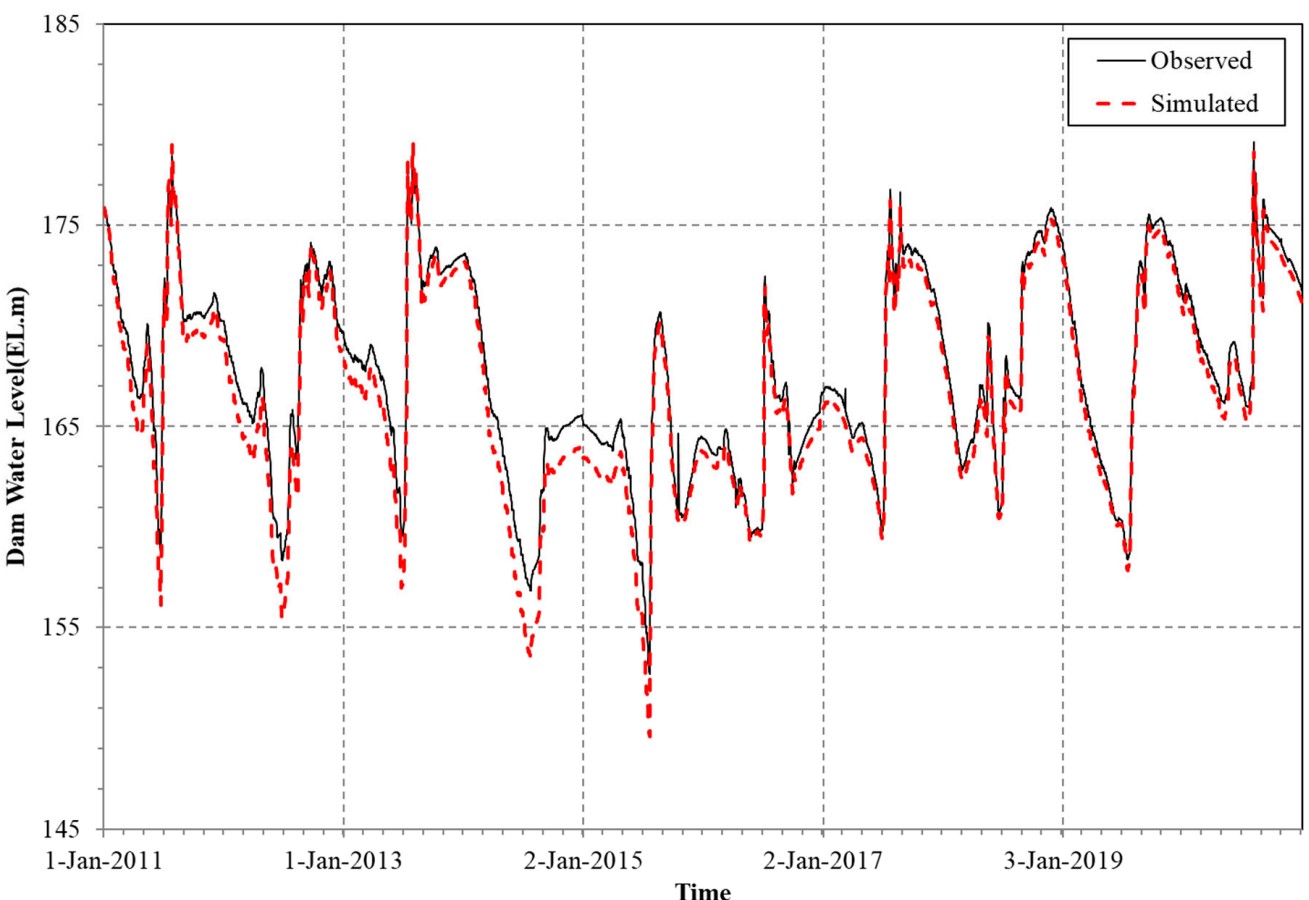

**Figure 2.** Evaluation results of HEC-5 (Hwacheon Dam).

*2.3. Study Area*

The simulation was applied to Hwacheon Dam, Chuncheon Dam, Uiam Dam, and Cheongpyeong Dam, which are hydropower reservoirs located in the Bukhangang watershed in South Korea. The dams are all connected in series, and the outflow discharge of the upstream dam affects the inflow of the downstream dam. These dams are located sequentially between the Pyeonghwa Dam located at the top of the Bukhan River and the Paldang Dam located at the point where the Bukhangang River joins the Han River, and the study area is shown in Figure 3.

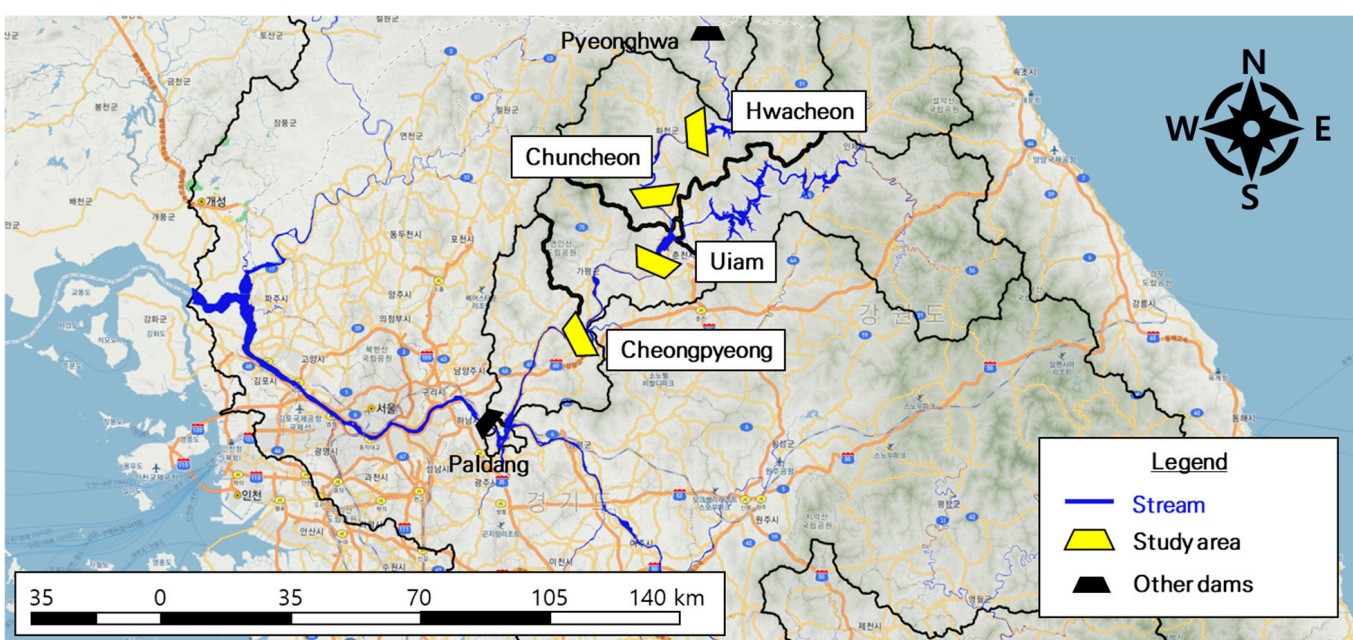

**Figure 3.** Study Area of Hydroelectric Dam. The trapezoidal symbols are hydroelectric dams located in the Bukhangang watershed. The yellow symbols are the target dams in this study. Pyeonghwa Dam is located at the most upstream.

The Hwacheon Dam is a gravity-type concrete dam with a height of 81.5 m and a length of 435 m located on the Bukhangang River in Hwacheon-gun, Gangwon-do, South Korea. It is the largest hydropower reservoir in Korea and is known to have water supply and flood control capabilities, unlike other hydropower reservoirs. It operates with a limit water level of EL.175 m during the flood season and a regular bay water level of EL.181 m and a low water level of EL.156.8 m during the non-flood season. Table 1 shows the specifications of other dams.

**Table 1.** The specifications of dams in study area.

| Dam | Year | Height (m) | Length (m) | Total Water Capacity (10⁶ m³) | NWL (EL.m) | LWL (EL.m) |
|---|---|---|---|---|---|---|
| Hwacheon | 1944 | 81.5 | 435 | 1018 | 181.0 | 156.8 |
| Chuncheon | 1964 | 40 | 453 | 150 | 103.0 | 98.0 |
| Uiam | 1967 | 23 | 273 | 80 | 71.5 | 66.3 |
| Cheongpyeong | 1943 | 31 | 407 | 185.5 | 51.0 | 46.0 |

*2.4. Assessment of Hydropower Resilience*

The resilience of a hydropower reservoir can be calculated at the dam water level, which is calculated from the inflow and outflow discharges and the current water storage. Since power generation is related to dam water level and outflow discharge, these are important considerations. If resilience is high, power generation can be increased, but there is a risk of water disaster. Therefore, to suggest the optimal operation rule using resilience, the methodology should be constructed by additionally considering spillway discharge, flood risk days, and drought risk days (Figure 4). Figure 4 shows the procedure for evaluating the power production performance of the hydropower reservoir.

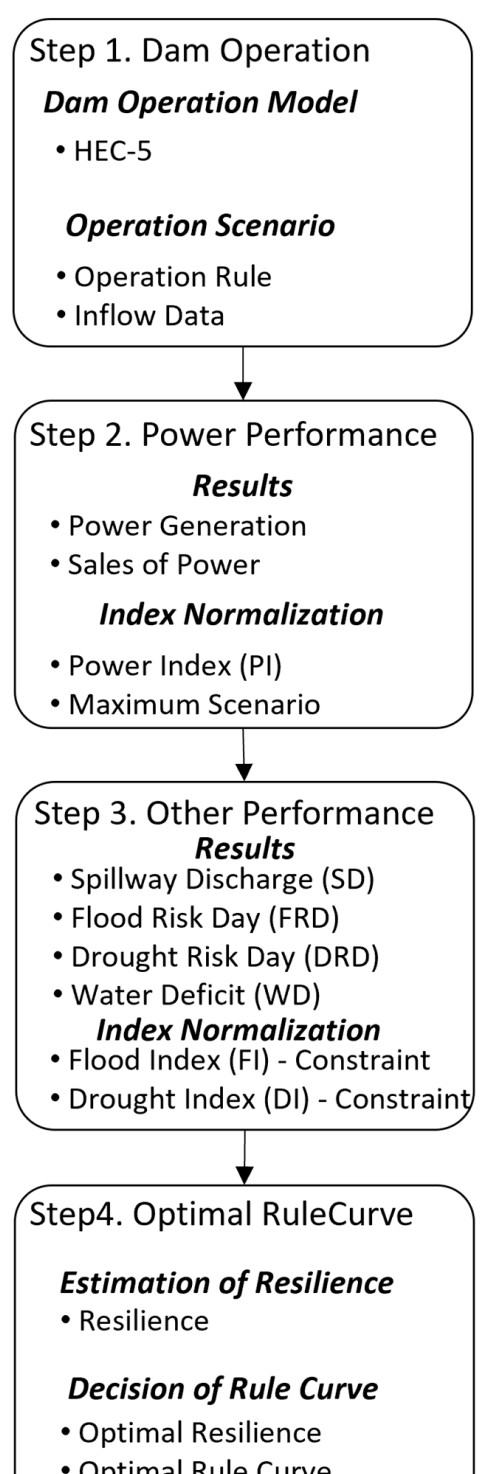

**Figure 4.** Assessment Methodology of Resilience.

The first step is to define the operating rules for inflow and hydropower reservoirs. For inflow, there is a method using historical data and a method using predicted inflow using machine learning such as LSTM (Long Short-Term Memory), which is one of the recurrent neural network techniques. However, in this study, only historical inflow data was used. The hydropower reservoir operation rules in Korea can be stipulated as limited water level, outflow discharge by water level, outflow discharge by period, and so on. For this standard, the relevant laws, and regulations, such as the regulation on the operation

of dams and weirs, were referred to [16]. The first step is to simulate the operation of the hydropower reservoir using the above data as input values. As the simulation results, dam water level, outflow discharge, power generation, etc. are calculated, and the program used for the simulation was HEC-5 developed by the US Army Corps of Engineers.

The second step is to evaluate the power generation, which is the main performance of the hydropower reservoir. However, the operation rule was defined as maximizing the amount of electricity sales rather than maximizing the amount of power generation. As shown in Figure 5, it was confirmed that the month with the highest dam water level had the lowest SMP (System Marginal Price) unit price from August to October. SMP means the most expensive price among power sources required to meet the demand for power. All power sources in Korea received the same SMP in return for power generation. There is a possibility that even if the power generation is the most in August and October when the dam water level is high, the power sales may not be the maximum. Therefore, in this study, not the maximum power generation, but the maximum electricity sales amount was defined as the power generation performance and an evaluation methodology was presented.

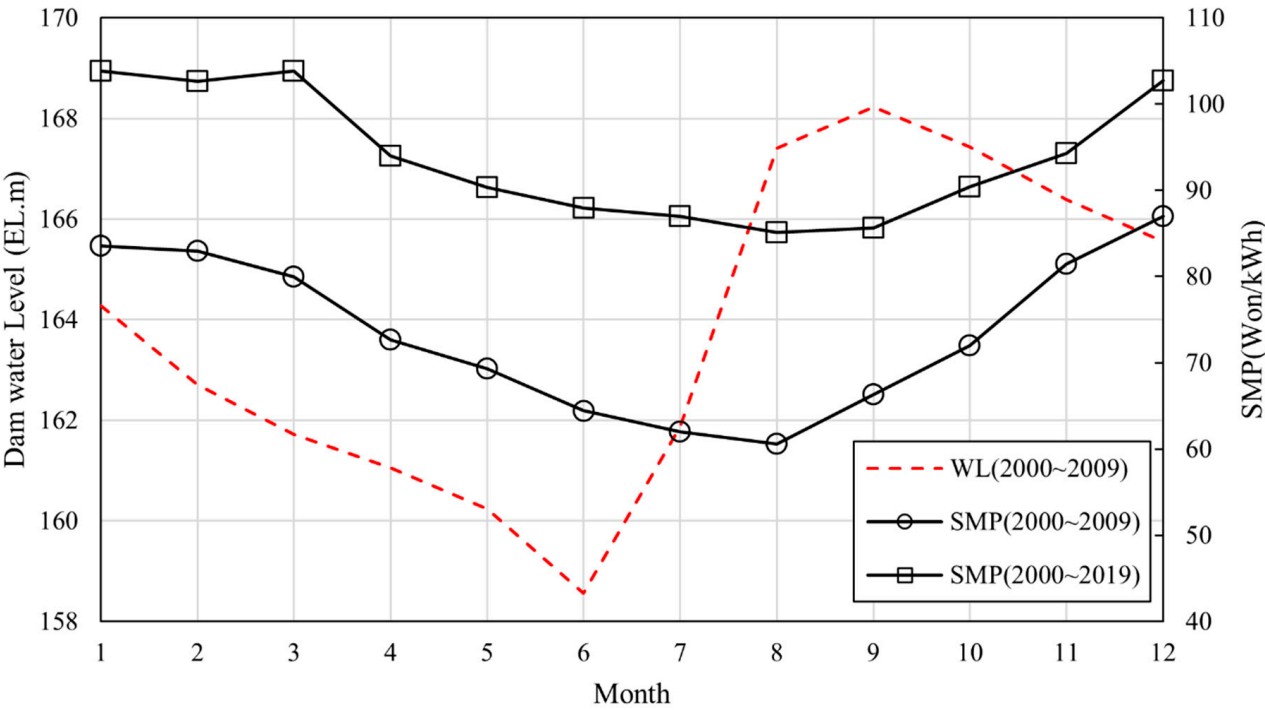

**Figure 5.** Comparison of monthly SMP price and water level at reservoir (2000~2019).

The third step is to evaluate the additional performances of the hydropower reservoir. Additional features are the performance of flood control and water usage. In the second stage, the optimal operation rule is selected based on the electricity sales, but if the power generation performances are similar, the optimal operation rule is selected based on the additional performances. In addition, if excessive storage and discharge are performed to maximize electricity sales, severe floods and droughts may occur. To prevent this, the limit value was defined as the possibility of inducing flood and drought. The fourth step is to evaluate the resilience of each scenario with Equation (1), quantify each performance by scenario, and determine the comparative advantage.

Basically, each performance is evaluated through simulation results. The power generation performance is calculated by Equation (2) using the power generation amount and monthly power sales unit price, and this value is normalized to 0~1 using the maximum and minimum values. Flood risk days (FSD) and spillway discharge (SD) were used for flood control performance as shown in Equation (3). For water usage performance, Drought Risk Day (DRD) and Water Deficit (WD) were used as in Equation (4), which means that

the current water level has dropped to Low Water Level (LWL), and it is impossible to proceed with discharge [17]. As with the power generation performance, since the values have different dimensions, they are normalized, and the range of values is converted to 0~1. The limit value was calculated based on the performance data of the past 10 years for flood control and water usage performance values. The equations for calculating each indicator are as follows.

$$PI = (PC - PC_{min})/(PC_{max} - PC_{min}) \tag{2}$$

$$FI = \left( \frac{FSD - FSD_{min}}{FSD_{max} - FSD_{min}} + \frac{SD - SD_{min}}{SD_{max} - SD_{min}} \right)/2 \tag{3}$$

$$DI = \left( \frac{DRD - DRD_{min}}{DRD_{max} - DRD_{min}} + \frac{WD - WD_{min}}{WD_{max} - WD_{min}} \right)/2 \tag{4}$$

where, $PI$ is the index for power generation performance, $FI$ is the index for flood control performance, $DI$ is the index for water usage performance. $PC$ is the electricity sales for each scenario, $FSD$ is the risk day of flood, $SD$ is annual spillway discharge, $DRD$ is the day of water usage, $WD$ is water deficit, $X_{max}$ is the maximum of each index, $X_{min}$ is the minimum of each index.

### 2.5. Application of Hydrologic Modeling

Daily data from 2006 to 2013 were used for the inflow, which has a great influence on the performance evaluation. In this period, sufficient time has elapsed since the inflow pattern changed due to the construction of Imnam Dam located upstream of Hwacheon Dam in 2003. Also, this period was before 2014–2015, when the severe drought occurred. To consider a general situation, not a disaster, a period with a relatively constant inflow pattern was set as a time interval.

The operating standards for hydropower reservoirs in Korea were investigated and reviewed to establish an operation scenario. The power generation method of the hydropower reservoir is a regular power generation type, but the outflow discharge according to the water level is not determined. However, since various dams are built in a small area, it is necessary to establish an operation plan every month in accordance with the River Act and the operation standards for dams and weirs. In the process of establishing this plan, after predicting the monthly inflow based on historical data, the monthly target water level and discharge plan are set. The basic principle is to prevent drought and flood damage from occurring, and the discharge plan is finally decided by referring to the electricity supply and demand plan and the power generation stop plan. The establishment of this plan determines the outflow discharge by predicting the inflow and setting the target water level. The target water level is generally selected based on June and September, before and after the flood season, and this value is also calculated as the average value of historical data. Therefore, in this study, a scenario was established in which the target water level and target discharge were adjusted and operated based on the monthly average water level and outflow discharge of each dam. Figure 6 is a diagram showing how to set the target water level and target discharge in Hwacheon Dam, and all are set to be adjusted at a certain rate based on the average value. Actually, the current dam operation is performed with the values indicated by the solid lines in Figure 6a,b. As shown in the dotted lines in Figure 6a, the scenario was set to increase or decrease the target water level at a certain rate. In addition, the outflow discharge in Figure 6b was also increased and decreased at a certain rate based on the average value to operate. Scenarios that correspond to both the scenario of changing the target water level and the scenario of changing the target discharge were set up as scenarios. As a result, both the target water level and discharge were divided into 15 stages between the lowest and highest values to construct a scenario. The total number of scenarios is 225.

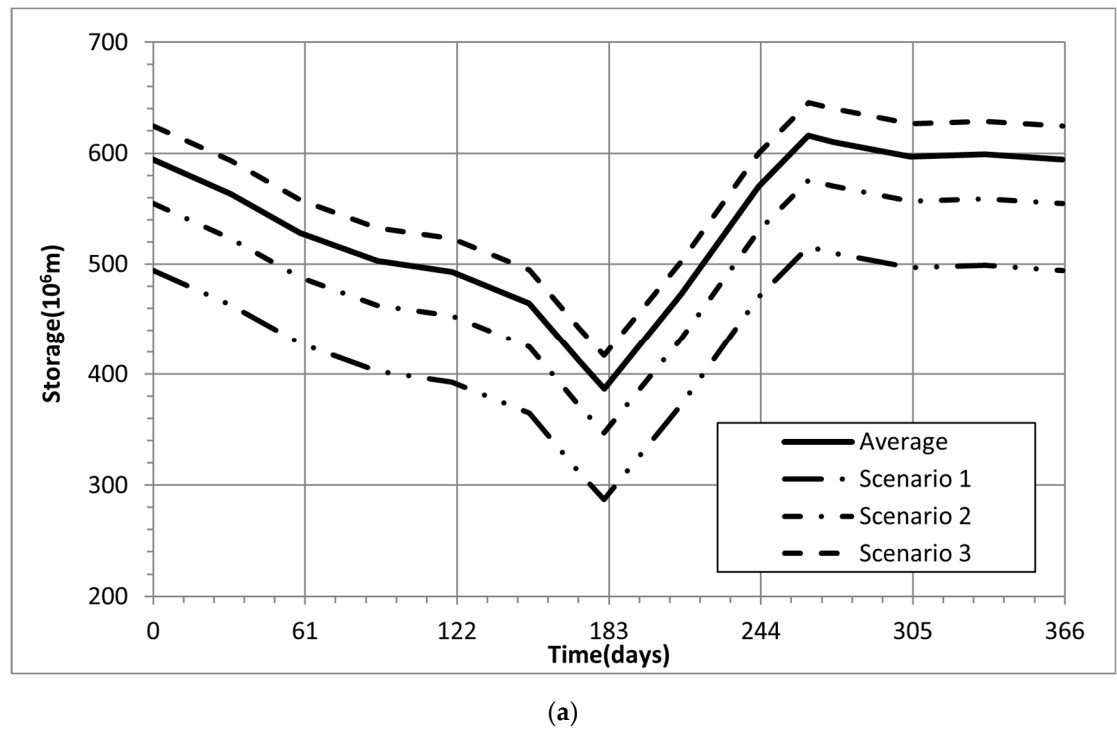

(**a**)

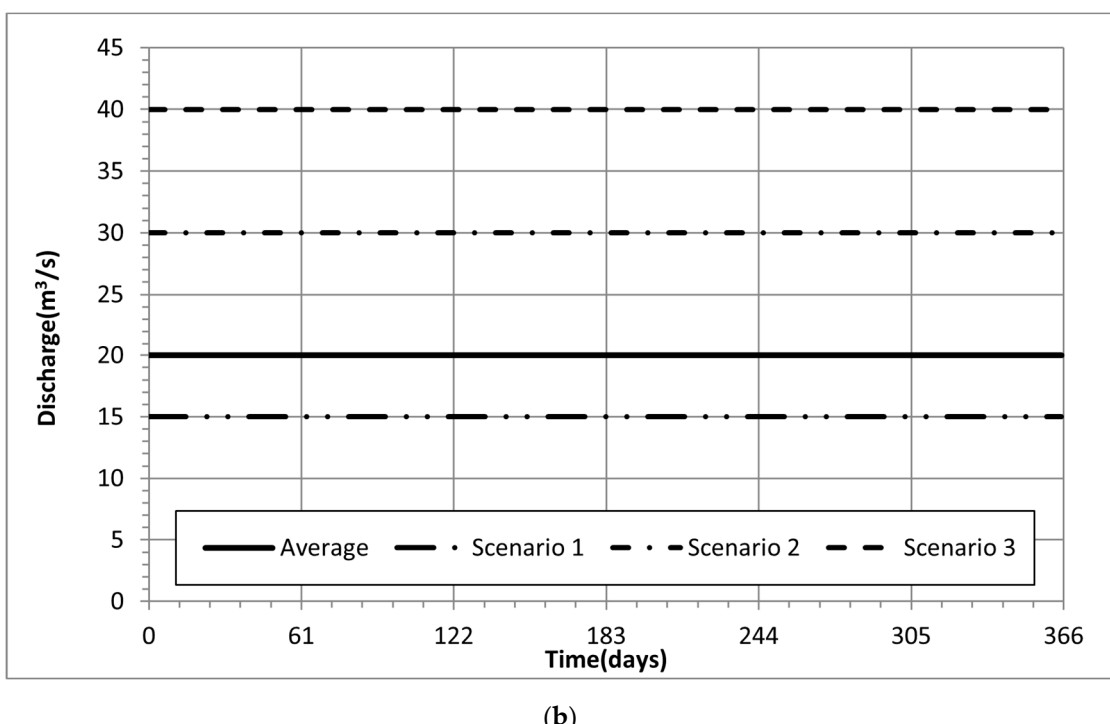

(**b**)

**Figure 6.** Scenario of dam operation at Hwacheon Dam. (**a**) Target of dam water level; (**b**) Target of outflow discharge.

## 3. Results of Dam Operation

### 3.1. Results of Historical Data

The concept of resilience was applied to the Hwacheon Dam using historical data. About 54% of the annual precipitation in Korea is concentrated during the flood season (June to September). Therefore, the hydropower reservoir in Korea was built for power generation, but since 1973, it has been contributing to flood control by setting a water level limit during the flood season. The limiting water level operation method causes losses in

terms of power generation. To confirm this, resilience was applied using historical data before and after 1973. Resilience was compared in 1981 and 1986, when the average annual inflow was like that of 1971, before the operation of the limited water level during the flood season. In Table 2, hydropower data, operation data, and resilience of hydropower reservoirs by year were calculated and presented. Figure 7 compares dam water levels by year. In Figure 7, in 1971, when there was no limiting water level during the flood season, the water level in the dam recovered the fastest to normal high-water level (NWL) (Point A). In 1981 and 1986, the water level of the dam was operated below EL.175 m during the flood season due to the limited water level (Point B & C), and the hydropower reservoir was operated by restoring the water level to NWL at the end of the flood season (Point D). As a result of applying the resilience defined in this study using historical data for the period, it was found that the resilience of 1971, when it was operated without a limiting level, was greater. There is a clear difference in resilience during the flood period.

**Table 2.** Comparison of hydrological data and resilience (All seasons are full year and the Flood season is from 21 June to 20 September).

| Contents | 1971 | 1981 | 1986 |
|---|---|---|---|
| Annual Mean Water Level (EL.m) | 171.6 | 171.1 | 169.3 |
| Annual Mean Inflow (m$^3$/s) | 100.91 | 116.64 | 98.13 |
| Annual Mean Outflow (m$^3$/s) | 88.84 | 81.82 | 85.35 |
| Spillway Discharge (m$^3$/s) | 6329 | 14,556 | 7666 |
| Power Generation (MWh) | 422,421 | 385,426 | 376,201 |
| Resilience (All Season) | 0.6100 | 0.5901 | 0.5153 |
| Resilience (Flood Season) | 0.7741 | 0.7464 | 0.5354 |

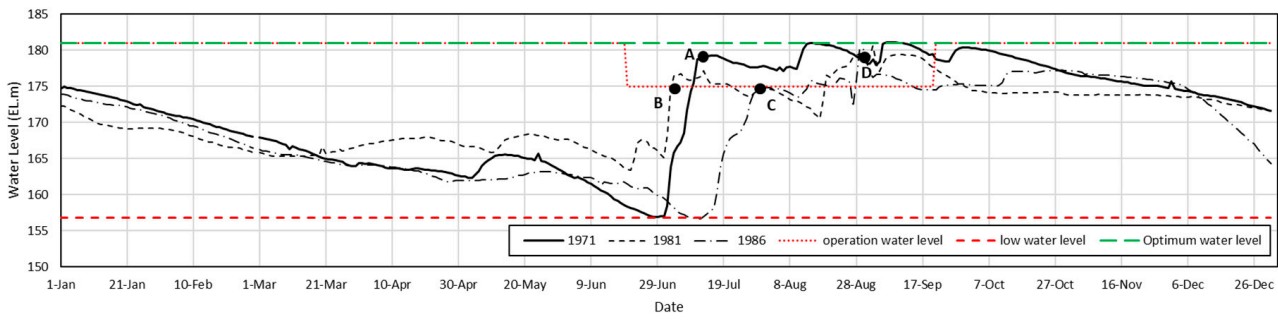

**Figure 7.** Comparison of dam water level by year.

Comparing generation, discharge, and resilience by year, we can confirm the importance of water level recovery in terms of power generation. Theoretically, to increase the amount of power generation, it is necessary to restore the dam level to NWL to secure an effective head. Therefore, in 1971, when the water level was restored the fastest, the resilience was the highest at 0.61 (all period). At this time, the annual total power generation amounted to 422,421 MWh in 1971, producing the largest amount of electricity. On the other hand, in 1981 and 1986, it was operated below the limit water level during the flood season, and the resilience was 0.59 and 0.51, respectively (Figure 8). Power generation also decreased to 385,426 MWh (91.2%) and 376,201 MWh (89.1%), respectively. The average annual outflow discharge in 1981 is less than in 1986, but the average annual generation is higher. The reason can be confirmed by the average annual dam water level. In 1981, it was confirmed that the water level recovered more rapidly during the flood period. In fact, when comparing the power generation in June, the power production in June 1981 was 38,547 MWh, which was almost twice as high as 20,826 MWh in 1986. Through this, it can be said that the resilience defined by the dam water level is related to the power generation

in hydropower plants. As shown in Figure 8, operation results in 1971, which had high resilience regardless of period, produced more electricity than results in 1981 and 1986. In addition, results in 1981 was mere resilient than results in 1986 and actually produced more electricity. Therefore, resilient operations can increase electricity production. To minimize the power loss in the multi-functional operation of the hydropower reservoir, it is necessary to operate the dam from the perspective of restoring the appropriate water level for power generation, such as resilience.

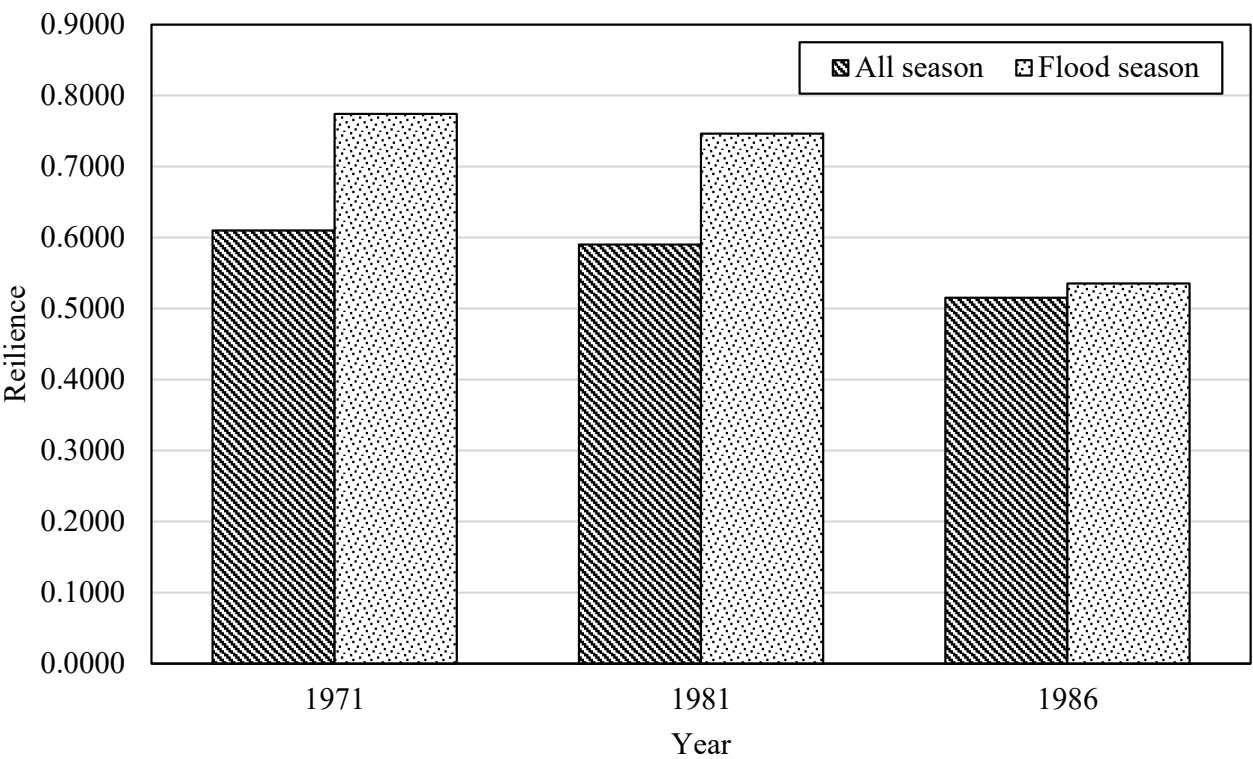

**Figure 8.** Comparison of resilience by year.

*3.2. Results of Simulation Data*

The specifications of the hydropower reservoir used in the simulation were provided by Korea Hydro & Nuclear Power, which operates them. Using the simulation results, a scenario in which the amount of electricity sales is maximized was derived, and then the additional performance was evaluated. After calculating the index values for each performance, the resilience of each scenario was finally calculated.

The simulated results by changing the target dam water level and outflow discharge were expressed in a matrix form as shown in Figure 9, and the optimal operation plan was derived using this. For each matrix type shown in Figure 9, the horizontal axis is the target water level change, and the vertical axis is the target outflow discharge. That is, it means that the target water level is adjusted upward as it goes to the right in the matrix, and the target outflow discharge increases as it goes down. It is a matrix with values ranging from 0 to 1, and the closer to 1, the better the performance. In the case of Hwacheon Dam, it was found that the method of lowering the target water level and increasing the outflow discharge can derive the largest amount of electricity sales. Figure 9b shows that as flood control performance increases outflow discharge and lowers the target water level, the better the performance. Conversely, as shown in Figure 9c, the water usage performance was found to be better as the target water level was adjusted upward while maintaining the target outflow discharge properly. As a result of the Hwacheon Dam, reducing the target water level and increasing the target outflow discharge resulted in the maximum electricity sales.

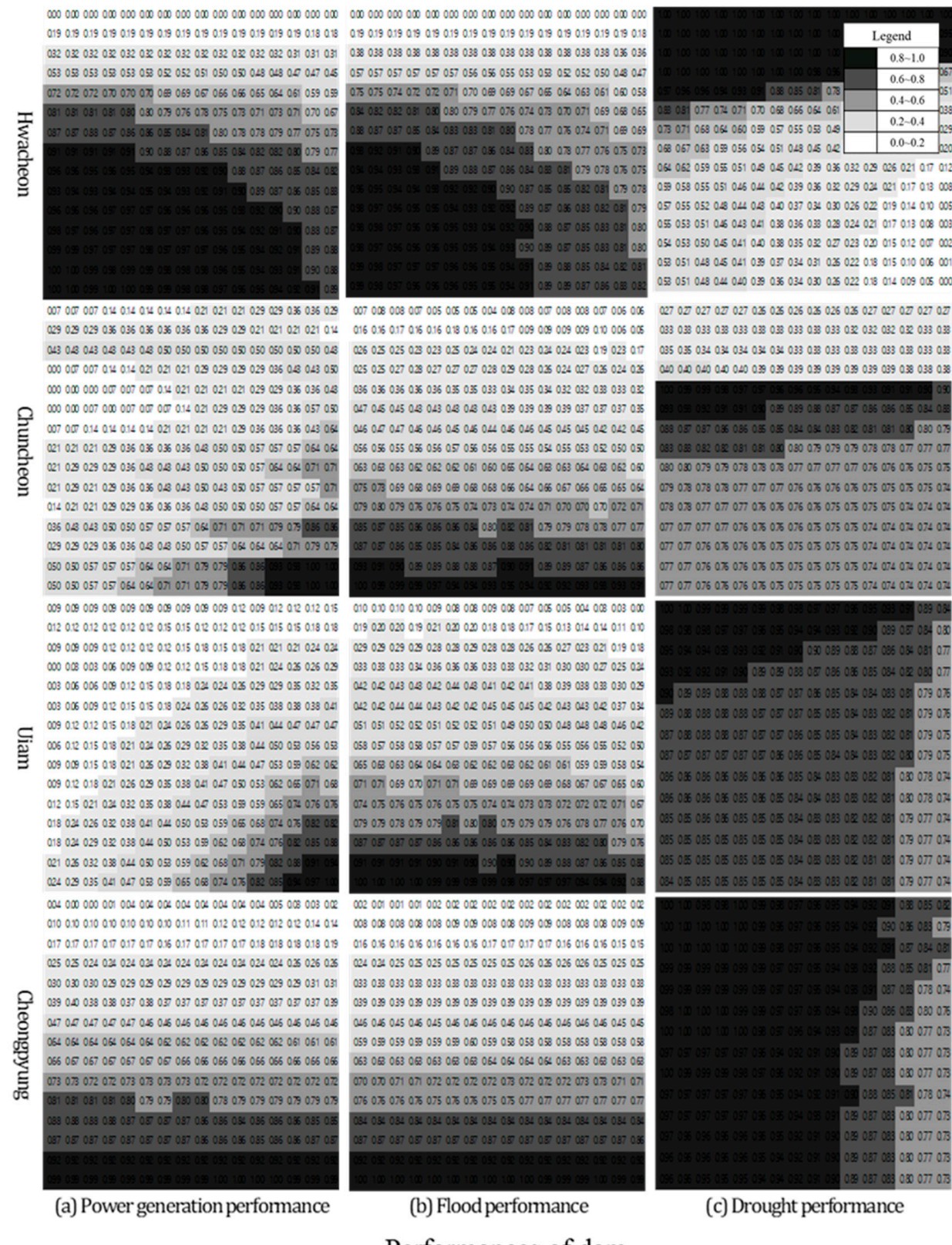

**Figure 9.** Performance matrix for each hydropower reservoir. (**a**) is the matrix calculated by Equation (2) as an evaluation of the power generation performance. (**b**) is the matrix calculated by Equation (3) as and evaluates the flood control ability. (**c**) is the matrix calculated by Equation (4) and evaluates the drought control ability.

Table 3 shows the scenario conditions in which the maximum electricity sales for each dam occurred and the performance results under those conditions. The resilience was calculated by Equation (1), and the additional performances are the results calculated by Equations (2)–(4). In the case of individual operation, it is advantageous to maximize the target water level and target outflow discharge, but in the case of linked operation, the operation should be carried out in consideration of the situation of the downstream dam. When electricity sales are operated to the maximum, the Uiam dam and Cheongpyeong dam perform better operations in terms of resilience, and in addition, flood control performance and water usage performance can be secured. In the case of the Hwacheon dam and Chuncheon dam, power generation can be secured by increasing outflow discharge, but water usage performance is low due to the low target water level. It will perform operations with low resilience. These results derived the maximum electricity sales within a given scenario with the entire hydropower reservoir system (4 dams). These were operated so that the increase in the outflow discharge of Hwacheon dam and Chuncheon dam located upstream kept the water level in Uiam dam and Cheongpyeong dam high and the amount of power generation increased.

**Table 3.** Performance Results of Hydropower Dam with Scenario.

| Dam | Scenario (WaterLevel/Outflow) | Power Generation (MWh) | Flood Performance | Drought Performance | Resilience |
| --- | --- | --- | --- | --- | --- |
| Hwacheon | −120,000/+30 | 204,942 | 0.94 | 0.64 | 0.350 |
| Chuncheon | +20,000/+60 | 120,451 | 0.93 | 0.74 | 0.300 |
| Uiam | +10,000/+60 | 174,531 | 0.92 | 0.77 | 0.729 |
| Cheongpyeong | −70,000/+60 | 416,556 | 1.00 | 0.94 | 0.858 |

## 4. Discussion

The results of the derived optimal scenario were compared with the historical data for 2006~2013. The optimal scenario was selected as the one with the largest amount of electricity sales. Because the unit price of electricity in Korea changes every day, the maximum value of production does not lead to the maximum value of sales amount. Therefore, the electricity sales were set as the objective function to consider the economic aspect. The electricity sales for each scenario were calculated by multiplying the time series of electricity production calculated in HEC-5 by the average monthly sales unit price in the past. Figure 10 shows the comparison between the historical data of Hwacheon Dam and the simulation results. As shown in Table 4, it was confirmed that the Hwacheon Dam operated at a lower target water level than the previous data. In addition, since the target outflow discharge was operated at a high level, it was confirmed that a larger amount was discharged than the historical data.

It was checked to see how effective the optimal operation results were compared to the past performance. The period for comparison is 2006–2013. Table 4 shows the results of comparing the power generation of all dams, which are operational goals, with the past performance. Hwacheon Dam and Chuncheon Dam, which were evaluated for their low resilience, produced less electricity than their past performance. However, the Uiam Dam and Cheongpyeong Dam, located downstream, produced more electricity than the historical data because a large amount of discharge was performed at a high dam water level. At this time, it was confirmed that the value of resilience was also highly evaluated. In the case of Cheongpyeong Dam, which is located the most downstream, about 50% more electricity could be produced. If the four dams were evaluated as a single power system, the power generation could be increased by 151,652 MWh. This is a result of an increase of about 19.83%. The operation of the hydropower reservoir considering resilience resulted in improved power production, which is the main performance. However, in this study, it cannot be said that all possible operating rules are reflected because it is simply a scenario

in which the target water level and outflow discharge are increased and decreased at a certain rate. Therefore, if research on random scenarios is added in the future, it is expected that the optimal operating rules for maximizing electricity sales will be obtained.

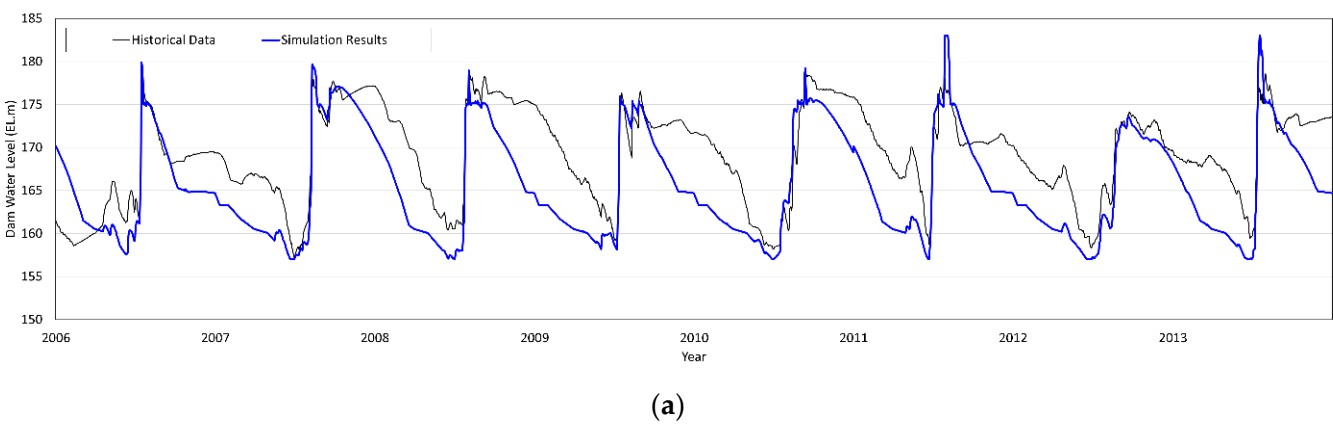

(**a**)

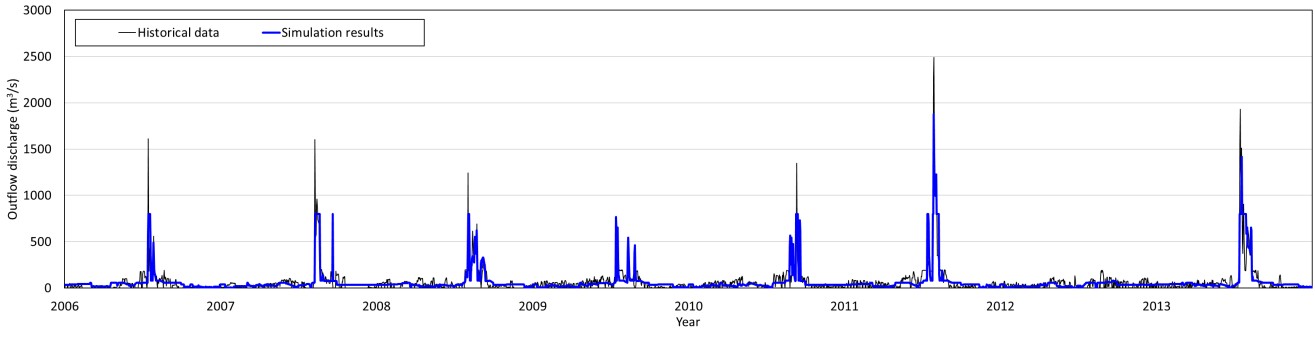

(**b**)

**Figure 10.** Comparison of historical data and simulation results. (**a**) Water level; (**b**) Outflow discharge.

**Table 4.** Comparison of historical data and simulation results for power generation.

| Dam | Power Generation (MWh) | | Comparison | Resilience |
|---|---|---|---|---|
| | **Historical Data** | **Simulation Results** | | |
| Hwacheon | 217,591 | 204,942 | −12,649 | 0.350 |
| Chuncheon | 130,571 | 120,451 | −10,120 | 0.300 |
| Uiam | 146,786 | 174,531 | 27,745 | 0.729 |
| Cheongpyeong | 269,880 | 416,556 | 146,676 | 0.858 |
| Total System | 764,828 | 916,480 | 151,652 | - |

## 5. Summary and Conclusions

From a recent policy perspective, the hydropower reservoir is a major component of the integrated water management system, and its status and value need to be re-evaluated as renewable energy for the realization of zero carbon. However, despite the progress in related technologies, the past methodologies are being used for dam operation and evaluation. In consideration of enhancing the status of the hydropower reservoir and strengthening its role as a water resource manager contributing to the water resource system, it was urgent to establish a plan to secure the efficiency of dam operation in terms of watershed management, including water supply and flood control. Therefore, in this

study, the concept of resilience was applied to the operation of a hydropower reservoir, and a methodology to evaluate it was presented. The goal of this study is to propose the optimal operation rules for hydropower reservoirs that can maximize power sales and secure flood control and water usage performance from a resilience point of view.

In this study, the concept of resilience suggested by Kim et al. (2021) was introduced and defined in the power generation system of a hydropower reservoir. The concept was applied to four hydropower reservoirs connected in series to the Bukhangang watershed in Korea. The operation scenario was constructed with the target water level and target outflow discharge set as variables when the dam manager establishes the multiple operation plan in Korea. Based on the historical data on dam water level and outflow discharge, a scenario was constructed by increasing or decreasing them at a certain rate. The optimal scenario was derived by setting the electricity sales as a target function. For the simulation, HEC-5 developed by the U.S. Army Corps of Engineers was used, and the simulation results were used to evaluate the power generation performance, flood control performance, and water usage performance. Power generation performance is evaluated by electricity sales. The flood control performance is calculated by the number of days at risk of flooding and spillway discharge, and the water usage performance is calculated by the number of days at risk of drought and the amount of water shortage. For the optimal operation rule, the power generation performance is prioritized, and the additional performance is limited to within the score calculated based on past performance data.

The optimal rule curve considering resilience was presented using the observation inflow data from 2006 to 2013. When operating hydropower reservoirs in connection, it was confirmed that they must be operated with high resilience to maximize power production. Comparing the simulation results with the past performance during 2006~2013, power generation increased by about 19.83% when operated with high resilience. However, since the current scenario consists of increasing or decreasing only at a certain rate, it cannot be considered that all possible scenarios are reflected. Therefore, if these limitations are overcome in the future, it will be possible to derive an operation rule that can secure additional performances while maximizing the electricity sales of the hydropower reservoir. It is expected to be able to present a methodology for rationally calculating an operational rule that improves resilience. In addition, the proposed methodology presents the flood control effect and the water usage effect as normalized values for relative comparison by scenario. In the future, if the flood control benefit and water supply benefit are calculated based on these values, the additional economic effect compared to the power loss can be quantified.

**Author Contributions:** Conceptualization, H.-J.S. and S.O.L.; methodology, D.H.K.; software, D.H.K.; validation, S.O.L.; formal analysis, T.L.; investigation, D.H.K.; resources, D.H.K.; data curation, S.O.L.; writing—original draft preparation, D.H.K.; writing—review and editing, T.L.; visualization, T.L.; supervision, S.O.L.; project administration, H.-J.S.; funding acquisition, H.-J.S. All authors have read and agreed to the published version of the manuscript.

**Funding:** This research was supported by KOREA HYDRO & NUCLEAR POWER CO., LTD (No.2019-RFP-New&Renewable-1) and Basic Science Research Program through the National Research Foundation of Korea (NRF) funded by the Ministry of Science, ICT & Future Planning (No.2021R1A2C2013158).

**Institutional Review Board Statement:** Not applicable.

**Informed Consent Statement:** Not applicable.

**Data Availability Statement:** Data sharing is not applicable to this article.

**Conflicts of Interest:** The authors declare no conflict of interest.

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
