# Peer review of "Generating More Hydroelecticity While Ensuring the Safety: Resilience Assessment Study for Bukhangang Watershed in South Korea"

_applsci, doi:10.3390/app12094583_

Round 1

Reviewer 1 Report

Dear Authors,

The submitted manuscript named "Generating more hydroelectricity while ensuring the safety : resilience assessment study for Hangang watershed in South Korea" has a clear and self-explanatory title. The topic of the submitted manuscript fits the scope of the journal and it is very interesting and of high importance to be reproduced. However some parts of your valuable research are not explicitly presented and analyzed. My overall understanding would be much better in case the materials and methods section would be more explicit and clarified. I would propose you present more on your background information you have for your case-study watershed (parameters, definitions used, assumptions made) since many of these parts are currently missing. This will also help the clarity of your manuscript and ease the reader while landing on the presentation of the results/charts/figures. 

Below you can find my comments and suggestions to further improve it.

L4: Please correct to the word "electricity" in the title

Abstract

Your abstract should contain quantifiable parts of your results which is not the case for the moment. Please elaborate on this during your review.

1.Introduction

L47-50 there is a duplication in these sentences. Please review it accordingly

Overall comment: The authors have nicely introduced in the historical background of operating hydropower reservoirs in Korea, with a presentation Korean policy and its long terms goals. However, this part governed the whole introduction section with no references to existing cases adopting the idea of resilience in operating hydropower reservoirs and limited explanation and references on how close the adopted definition is applicable.

Please add more references on the integration of resilience analysis and assessment cases for hydropower/ flood mitigation plans. This will provide wider and better background while presenting your project outcomes.

Some propositions:

  • Ecological flows to sustain healthy environment in the reservoirs' drainage areas and how they affect your overall resilience of the reservoir operation (energy, flood control, water use)
  • The effects of use of pump storage and how you dealt with this in your case? Applied any kind of filters or pump storage is not applicable in your reservoirs' system?

2,Materials and methods

L147 there is a duplication in the reference please review it

L125-155 can be transferred back to introduction section

More information about the HEC-5 model is required to be presented to the readers

Which were your HEC-5 hydrological model evaluation criteria set to be so as to directly use the model outputs? Can you provide more information on the topic (e.g. NSE performance) ?

3.Study area

L210 - you mention that the hydropower reservoirs are located in the  "Bukhangang watershed in South Korea" while the title of your manuscript is referring to Hangang. Please review it.

In table 1 you miss a column with the total water capacity of the dams. You can also skip the address column, otherwise correct the typo.

L234 - define the term LSTM

L245 - define the term SMP

L264 - define the term LWL

The whole section 3 should be merged with section 2 (materials and methods) no need to have a section with this size of content.

4. Application Methodology

The whole section 4 should be merged with section 2 (materials and methods) no need to have a section with this size of content.

Apart from the use of equations 2,3 and 4 how did you end up with the final result of resilience? Equation 1 is not clear how it was used.

L296-297 please explain in the text how the scenarios are differentiating

Figure 5. please explain for which dam this is applicable 

5. Results of historical data

Results sections should be merged, meaning current sections 5 and 6, creating the new section 3. Why do you need two different sections (current 5 and 6) to present historical and simulation data, other than create just 2 separate paragraphs (new 3.1 and 3.2) ?

Figure 6 would be better if it was in color, otherwise the style must be changed. Also operation and low water level have the same styles selected. Please review accordingly.

Figure 7 doesn’t present a comparison of annual resiliencies other than reservoir elevation. Currently looks the same as Figure 6 but with filled colors.

6. Results of simulation data

L344-348 need to be better explained, it is difficult to follow you

Figure 8 - the legends of horizontal and vertical axes are missing

L356-357 how did you calculated the numbers for flood and performances in order to estimate the overall resilience per reservoir? Please elaborate on this on the text

7.Discussion

L371 -  how you define and quantify the term "optimal scenario" ? Please elaborate on the manuscript

L379-351 please clarify which periods of performance are compared

8. Summary and conclusions

Proper revisions need to be elaborated based on the review of the previous sections

Reviewer 2 Report

Throughout - Change Fig. to Figure and make sure all are capitalized. Change past to historical - unless there is a reason not to. If not, define past data. 

Line 49 - remove "played a role" and following sentence. Seems to be a typo/repeated sentence. 

Line 51 - developed/developing - replace one of these with a synonym

Line 87 - change second absorbed to "emitted"

Line 89/90 - remove or define "new" energy

Line 137 - be consistent with "engineering" - all capitalized or all lower case

Line 236 - generally avoid the use of etc. in academic writing. Remove or replace 

Round 2

Reviewer 1 Report

Dear Authors  thank you for providing your responses.